# Burden of Influenza and Respiratory Syncytial Viruses in Suspected COVID-19 Patients: A Cross-Sectional and Meta-Analysis Study

**DOI:** 10.3390/v15030665

**Published:** 2023-03-01

**Authors:** Vivaldo Gomes da Costa, Ana Júlia Chaves Gomes, Cíntia Bittar, Dayla Bott Geraldini, Pâmela Jóyce Previdelli da Conceição, Ágata Silva Cabral, Tamara Carvalho, Joice Matos Biselli, Paola Jocelan Scarin Provazzi, Guilherme Rodrigues Fernandes Campos, Paulo Ricardo da Silva Sanches, Paulo Inácio Costa, Maurício Lacerda Nogueira, João Pessoa Araujo, Fernando Rosado Spilki, Marília Freitas Calmon, Paula Rahal

**Affiliations:** 1Laboratório de Estudos Genômicos, Departamento de Biologia, Instituto de Biociências Letras e Ciências Exatas, Universidade Estadual Paulista Júlio de Mesquita Filho (UNESP), São José do Rio Preto 15054-000, SP, Brazil; 2Laboratório de Pesquisas em Virologia (LPV), Departamento de Doenças Dermatológicas, Infecciosas e Parasitárias, Faculdade de Medicina de São José do Rio Preto (FAMERP), São José do Rio Preto 15090-000, SP, Brazil; 3Laboratório de Virologia Molecular, Departamento de Ciências Biológicas, Faculdade de Ciências Farmacêuticas (UNESP), Araraquara 14800-903, SP, Brazil; 4Departamento de Análises Clínicas, Faculdade de Ciências Farmacêuticas (UNESP), Araraquara 14801-360, SP, Brazil; 5Instituto de Biotecnologia, Universidade Estadual Paulista Júlio de Mesquita Filho (UNESP), Botucatu 18607-440, SP, Brazil; 6Laboratório de Microbiologia Molecular, Instituto de Ciências da Saúde, Universidade Feevale, Novo Hamburgo 93525-075, RS, Brazil

**Keywords:** SARS-CoV-2-negative, non-COVID-19, respiratory syncytial virus, influenza virus, a systematic review

## Abstract

Non-SARS-CoV-2 respiratory viral infections, such as influenza virus (FluV) and human respiratory syncytial virus (RSV), have contributed considerably to the burden of infectious diseases in the non-COVID-19 era. While the rates of co-infection in SARS-CoV-2-positive group (SCPG) patients have been determined, the burden of other respiratory viruses in the SARS-CoV-2-negative group (SCNG) remains unclear. Here, we conducted a cross-sectional study (São José do Rio Preto county, Brazil), and we collected our data using a meta-analysis to evaluate the pooled prevalence of FluV and RSV among SCNG patients. Out of the 901 patients suspected of COVID-19, our molecular results showed positivity of FluV and RSV in the SCNG was 2% (15/733) and 0.27% (2/733), respectively. Co-infection with SARS-CoV-2 and FluV, or RSV, was identified in 1.7% of the patients (3/168). Following our meta-analysis, 28 studies were selected (*n* = 114,318 suspected COVID-19 patients), with a pooled prevalence of 4% (95% CI: 3–6) for FluV and 2% (95% CI: 1–3) for RSV among SCNG patients were observed. Interestingly, FluV positivity in the SCNG was four times higher (OR = 4, 95% CI: 3.6–5.4, *p* < 0.01) than in the SCPG. Similarly, RSV positivity was significantly associated with SCNG patients (OR = 2.9, 95% CI: 2–4, *p* < 0.01). For subgroup analysis, cold-like symptoms, including fever, cough, sore throat, headache, myalgia, diarrhea, and nausea/vomiting, were positively associated (*p* < 0.05) with the SCPG. In conclusion, these results show that the pooled prevalence of FluV and RSV were significantly higher in the SCNG than in the SCPG during the early phase of the COVID-19 pandemic.

## 1. Introduction

Acute respiratory viral infections have profoundly impacted the socioeconomic status and health of people throughout human history [1,2]. Severe acute respiratory syndrome coronavirus 2 (SARS-CoV-2), the etiologic agent of coronavirus disease 2019 (COVID-19), was a new outbreak that emerged in Wuhan city, China, in late December 2019 [3,4]. The World Health Organization declared COVID-19 a pandemic on 11 March 2020 [5], and since then, the number of COVID-19 cases has rapidly increased worldwide [6]. As of 28 February 2023, more than 675 million cases of COVID-19 have been confirmed worldwide, with more than 6.8 million cases of deaths [7]. Similarly, influenza viruses (FluV) have also played a prominent role in public health challenges, causing up to 650,000 seasonal flu-associated respiratory deaths annually and 4 pandemics over the past 100 years (1918/H1N1, 1957/H2N2, 1968/H3N2, and 2009/H1N1) [8,9,10]. Human orthopneumovirus (previously known as the respiratory syncytial virus (RSV)) is another serious respiratory pathogen that is frequently a cause of hospital admissions related to bronchitis, bronchiolitis, and pneumonia, particularly in children, seniors, or immunocompromised patients [11,12,13]. In 2019, RSV infection, including only ≥60-year-old adults in high-income countries, was estimated to encompass approximately 5 million cases, including 33,000 in-hospital deaths [14]. These findings show the impact of SARS-CoV-2, FluV, and RSV seasonality on the global burden of infectious diseases.

SARS-CoV-2 and RSV share a similar transmission route as that of FluV, with dominant transmission by inhalation via virions suspended in aerosols and/or droplets expelled from the respiratory tract after breathing, sneezing, coughing, or close physical contact with an infected individual. An alternative form of transmission, referred to as ‘indirect’, is via contact with contaminated surfaces (fomites) [15]. After initial infection, symptomatic cases progressing to COVID-19 may have an average incubation period of 5.2 days (95% CI, range: 4.1–7.0) [16]; the average incubation period for RSV infection is 3–5 days, while that for influenza is 1–4 days [12,17]. There are both similarities and differences between the clinical features of influenza, RSV infection, and COVID-19. Regarding clinical similarities, the three diseases cannot be differentiated by only signs and symptoms, including non-specific cold-like clinical symptoms such as fever, chills, cough, difficulty breathing, and bronchiolitis, which is highly frequent in RSV infections, fatigue, sore throat, runny or stuffy nose, muscle pain or body aches, headache, and vomiting. The clinical differences involved are related to several parameters, for instance, a higher frequency of diarrhea in children with flu. Although it can occur at all ages of COVID-19 cases, change in or loss of taste or smell is more frequent in COVID-19. The three viruses can progress to complications and severity, represented by SARS, although at present, this is more frequent with COVID-19 [18,19].

Although meta-analyses have evaluated the co-infection rate between SARS-CoV-2 and other respiratory viruses [20,21,22,23], no studies to date have analyzed the background prevalence and impact of other respiratory viruses in SARS-CoV-2-negative individuals. Indeed, the pooled results allow a better understanding and updating of the issue, in addition to providing greater robustness in identifying the differences between the studied populations. Here, we conducted a meta-analysis to determine the positivity of respiratory pathogens (i.e., FluV/RSV) in the SARS-CoV-2-negative group (SCNG). Additionally, in light of the knowledge gaps inherent to the under-reporting of non-SARS-CoV-2 respiratory viruses, we conducted a cross-sectional study and included our results in the pooled estimate.

## 2. Materials and Methods

### 2.1. Ethical Approval

This study was part of the Corona-ômica.BR/MCTI network, following Brazilian regulations and international ethical standards and was approved by the Research Ethics Committee (protocol number: 33202820.7.1001.5348).

### 2.2. Cross-Sectional Study, Specimen Collection, and Processing

From May to September 2020, clinical samples with a clinical suspicion of COVID-19 were collected by healthcare professionals and their close contacts in the municipality of São José do Rio Preto, Brazil. During this period, the definition of suspicion of the disease was based on the presence of symptoms such as fever or acute respiratory discomfort (nasal congestion, cough, sore throat, and shortness of breath), headache, loss or alteration of smell or taste, ocular symptoms, myalgia, arthralgia, fatigue, nausea, vomiting, and diarrhea. The demographic details collected included sex, age, date of onset of symptoms, and place of collection. Clinical samples consisted of oro-nasopharyngeal swabs soaked in 3 mL of 0.9% saline sent to the Laboratório de Estudos Genômicos, Unesp, São José do Rio Preto, for molecular diagnosis. Therefore, following RNA extraction with a purification system using magnetic beads [24], a quantitative reverse transcription–polymerase chain reaction (qRT-PCR) test was performed [25]. After diagnosing for SARS-CoV-2, aliquots of samples stored at −80 °C were retrospectively screened, according to the established protocol, using a multiplex qRT-PCR assay for a panel of respiratory viruses: Alphainfluenzavirus influenzae (FluAV), Betainfluenzavirus influenzae (FluBV), and RSV [26].

Descriptive statistics were used, and demographic variables (i.e., sex and age) were used for the analysis between the groups via the chi-squared test, with a *p*-value of <0.05.

### 2.3. Systematic Review and Meta-Analysis

We performed a systematic review and meta-analysis according to the Preferred Reporting Items for Systematic Reviews and Meta-Analysis (PRISMA) protocol (Appendix A) [27]. The study protocol is available in PROSPERO with the registration number CRD42022367248 and can be viewed at https://www.crd.york.ac.uk/prospero/display_record.php?ID=CRD42022367248, accessed on 9 February 2023.

#### 2.3.1. Search Strategy 

After defining the research protocol, we performed a systematic search of PubMed, SciELO, and Google Scholar. Peer-reviewed articles in English, Spanish, or Portuguese were screened, and a combination of descriptors was used: (“SARS-CoV-2”, OR “COVID-19”, OR “SARS-CoV-2 negative”, OR “non-SARS-CoV-2”, OR “COVID-19 negative”, OR “non-COVID-19”) AND (“non-respiratory viruses” OR “influenza virus”, OR “influenza”, OR “respiratory syncytial virus”, OR “RSV”). We also identified additional studies by screening the reference lists of the selected articles and highly cited reviews on the topic of interest.

The following inclusion criteria were used: (1) original articles published in scientific journals that contained information on molecular surveys for the detection of FluAV/FluBV and RSV in non-COVID-19 individuals during the COVID-19 pandemic; (2) studies containing data related to the proportion and rate of viral infection using laboratory tests; (3) demographic data from FluV- and RSV-positive individuals for the analysis of other factors such as sex, collection period, and age; and (4) all target studies published during the COVID-19 pandemic. The following exclusion criteria were implemented in our search strategy: (1) absence or confusing specification of the outcome of interest regarding FluV/RSV positivity in laboratory tests; (2) revisions, book chapters, and molecular and seroprevalence studies not involving humans; and (3) small-scale molecular epidemiology studies with a sample size <20.

#### 2.3.2. Data Analysis

For all the selected studies, various criteria such as demographic information (i.e., first author, year of publication, place of study, and baseline characteristics of study participants including mean age, sex percentage, method of diagnosis, the proportion of positive humans investigated for respiratory virus infection, and clinical signs or symptoms) were used to analyze the data. The Newcastle–Ottawa Scale (NOS) was used to evaluate the risk of bias in the included studies [28]. The NOS evaluates group comparability and study group selection and is used to verify the exposure or outcome of interest in observational studies. NOS components were summed, and final scores were classified as follows: 0–2 low quality, 3 moderate quality, and 4–5 high quality [28].

#### 2.3.3. Statistical Analyses

Data were extracted using Microsoft Excel. Several tables were generated comprising dichotomous data (presence or absence of respiratory viruses infection) for the relative and cumulative calculation of the frequencies of the outcomes of interest, with 95% confidence intervals (CIs) calculated whenever possible. Meta-analysis was conducted using STATA IC/64 version 13.1 software (Stata Corporation, College Station, TX, USA) using the metaprop, metafunnel, and metaninf commands. The prevalences of FluV, RSV, and SARS-CoV-2 were determined by dividing the confirmed number of cases by the pooled denominator, with the results expressed as percentages. The variance of each frequency estimate (known as ES (effect sizes)) was calculated as pq/n, where p is the frequency, q is 1−p, and n is the total number of individuals screened [29]. The CIs for the average ES were calculated using the formula 95% CI = ES ± 1.96 × SE, where SE is the standard error (SE = √(pq/n)). To ensure proportionate weight distribution to the studies presenting extreme frequency (near 0 or 1), we applied the Freeman–Tukey arcsine methodology [30,31,32]. Heterogeneity, that is, any kind of variability among the results of the studies, was assessed using the I^2^ index and Cochran’s Q test. I^2^ values of 25%, 50%, and >75% indicated low, moderate, and high heterogeneities, respectively, among the studies. Cochran’s Q test *p*-value  <  0.05 was consistent with significant heterogeneity (I^2^ > 75%) [32]. Due to the nature of the studies, the existence of heterogeneity was expected; therefore, we chose to use the random-effect model for the meta-analysis as proposed by DerSimonian and Laird [33]. We performed a sensitivity analysis to test the effect of the influence of each study on the overall estimate. Furthermore, we conducted a subgroup analysis to reduce the possibility of heterogeneity. Publication bias was determined through a visual inspection of Begg’s funnel plot as well as with Egger’s test calculations. For all tests, *p*-values < 0.05 were considered to be statistically significant [34,35].

## 3. Results

### 3.1. Cross-Sectional Study

A total of 901 suspected COVID-19 cases were tested for SARS-CoV-2 and other respiratory viruses, such as FluV and RSV. Among all the individuals, there was a female predominance when stratifying by sex, with a female-to-male ratio of 0.57. The participants ranged from 1 to 92 years old (mean age, 35.5 years (±17.5 standard deviation (SD)). Initially, the molecular diagnosis of SARS-CoV-2 was prospective, and 18.6% (168/901) patients were confirmed as COVID-19-positive using qRT-PCR. The mean number of days between the onset of symptoms and sample collection from suspected COVID-19 patients for qRT-PCR was 3.6  ±  5 (SD) and 3.55 ± 5 (SD) days of illness for negative PCR and positive PCR, respectively. In relation to gender, the incidence of SARS-CoV-2 was 53.3% (88/168) among females (*p* = 0.2). Among the 10–19-year-old patients, there were minor odds of SARS-CoV-2 positivity (*p* < 0.01) than other age-stratified subgroups (Table 1). Similarly, among the samples collected on June/2020, there were minor odds of SARS-CoV-2 positivity (*p* < 0.05).

Regarding the SCNG, 1.9% (14/733) and 0.13% (1/733) tested positive for FluAV and FluBV, respectively. The FluV-positive group comprised 76% (13/17) females (*p* = 0.1). Co-infection with SARS-CoV-2 and FluV was identified in 1.78% of the patients (3/168). FluV positivity did not significantly increase between the age-stratified subgroups (*p* > 0.05); however, among the samples collected in July 2020, there were higher odds of FluV positivity (*p* < 0.001) (Table 1).

Regarding the SCNG, 0.27% (2/733) tested positive for RSV. Co-infection with SARS-CoV-2 and RSV was identified in 1.78% of the patients (3/168). The RSV-positive group comprised 80% (4/5) females (*p* = 0.3). RSV positivity did not significantly increase between the age-stratified subgroups or biological sample collection months (Table 1).

### 3.2. Systematic Review and Meta-Analysis: Characteristics of the Included Studies

We found 1143 reference articles in the electronic database using the descriptor combination established in the “Search Strategy” section. After the removal of duplicates (401), the title and abstract of the remaining records were screened for eligibility (681). Of these, 61 records were selected for reading their full text. As a result, 27 articles that met the eligibility criteria were included in this meta-analysis [36,37,38,39,40,41,42,43,44,45,46,47,48,49,50,51,52,53,54,55,56,57,58,59,60,61,62]. A flowchart of this selection step is shown in the Appendix A.

The 27 articles selected, in addition to the data from our cross-sectional study, reported a total of 114,318 suspected COVID-19 patients, the majority of whom were from the USA (48%) [37,39,40,43,45,46,48,50,53,54,55,61], followed by Brazil (15%) [44,47,48,49], Taiwan (7%) [42,62], Canada (4%) [52], England (4%) [56], Bangladesh (4%) [47], India (4%) [60], Japan (4%) [41], Israel (4%) [59], Turkey (4%) [36], and Spain (4%) [38]. The studies were conducted between early 2020 and April 2021. The age range of these patients was considerably heterogeneous, ranging from <1 to 106 years. The difference in sex between females and males was not distinct (Appendix A). Regarding the study quality evaluated using the NOS, bias was considered unlikely, as the mean score attained was moderate (four scores: Appendix A).

### 3.3. Clinical Features in the SCNG and SARS-CoV-Positive Group (SCPG)

Epidemiological trends regarding the clinical characteristics of the SCNG versus the SCPG showed that fever, cough, sore throat, headache, myalgia, diarrhea, and nausea/vomiting were positively associated with the SCPG. In contrast, rhinorrhea was positively associated with SCNG. Difficulty breathing and abdominal pain showed no significant association between the two groups (Figure 1) [42,46,47,51,54,59].

### 3.4. A Meta-Analysis to Estimate the Pooled Prevalence of FluAV/FluBV and RSV in the SCNG

As a result of laboratory confirmation via nucleic acid amplification, the estimated pooled prevalence of FluV was 4% (95% CI: 3–6), while for the analysis based on FluAV and FluBV, the results were 3% (95% CI: 2–5) and 1% (95% CI: 0–2%), respectively (Figure 2). For FluAV subtypes, the pooled positivity in relation to H1N1-pdm09 was 2% (95% CI: 0–4), while that for H3N2 was 2% (95% CI: 0–12) [37,38,39,40,41,42,43,44,45,46,47,48,49,50,52,53,54,55,56,57,58,59,60] (Appendix A).

For RSV, the estimated pooled prevalence was 2% (95% CI: 1–3) in the SCNG (Figure 3) [36,37,38,39,40,41,42,43,44,45,46,48,50,51,52,53,54,55,57,58,59,60,61,62]. Regarding the two major antigenic groups of RSV, A and B, there was a similar prevalence between RSV-A and -B, with a pooled rate of 1% (95% CI: 0–5) (Appendix A).

The prevalences of FluV and RSV in relation to the geographical origin of the studies are shown in Figure 4.

### 3.5. FluV and RSV Co-Infection: Epidemic Trends of FluV and RSV Infection

The pooled prevalence of FluV in the SCPG (i.e., co-infection) was 0% (95% CI: 0–1). Similarly, the pooled prevalence of RSV in the SCPG was 0%. Indeed, this group of patients had lower odds of FluV positivity (OR = 0.22; 95% CI: 0.18–0.27; *p* < 0.0001) and RSV positivity (OR = 0.33; 95% CI: 0.23–0.47; *p* < 0.0001) than the SCNG [38,39,40,41,43,44,45,46,47,48,49,50,52,53,54,55,56,57,58,59,61,62].

The epidemiological trends regarding the sample collection period showed little difference in FluV positivity, as the positive rate for the first half of 2020 was 4% (95% CI: 2–6) [36,37,38,39,41,42,43,44,45,46,52,53,54,55,56,61] and for later collections of 2020 was 5% (95% CI: 2–10) [40,47,48,49,50,57,59,60,62]. For RSV, a similar prevalence was observed for the samples collected in the first six months of the pandemic in relation to those collected in later periods (i.e., 3%, 95% CI: 1–5 vs. 1%, 95% CI: 0–2, respectively) (Appendix A).

### 3.6. Publication Bias and Sensitivity Analysis

For the analysis of publication bias, the funnel plot indicated that the existence of a selection bias was unlikely in relation to FluV, FluBV, and sample sizes among SCNG patients (Appendix A). Furthermore, using Egger’s test, we also found no sign of publication bias (i.e., bias coefficient > 0.05). However, publication bias was only observed in studies that estimated FluAV and RSV positivity among SCNG patients (*p* < 0.05). Four studies were observed to have zero viral positivity contributing to this outcome [41,51,57,60]. Regarding the sensitivity analysis, to assess each individual study on the combined viral prevalence via the removal of individual studies, no study significantly affected the combined positivity (Appendix A).

## 4. Discussion

This cross-sectional study and meta-analysis showed the prevalence of FluV and RSV in patients with a clinical suspicion of COVID-19 in 2020/mid-2021 [36,37,38,39,40,41,42,43,44,45,46,47,48,49,50,51,52,53,54,55,56,57,58,59,60,61,62]. Interestingly, the pooled prevalence of 4% for FluV and 2% for RSV in the SCNG was significantly higher than in the co-infection group (i.e., FluV/RSV positivity plus SARS-CoV-2), which had a pooled prevalence of 0% (*p* < 0.001). Compared with non-COVID-19 patients, cold-like symptoms were more positively associated (*p* < 0.05) with confirmed cases of COVID-19, including fever, cough, sore throat, headache, myalgia, diarrhea, and nausea/vomiting. Collectively, our study provides evidence of the early stages of the COVID-19 pandemic.

The retrospective detection of FluV and RSV, as well as other pathogenic agents, is necessary because, during the initial phase of the pandemic, all efforts were focused on the detection of SARS-CoV-2. Even though the 2567 reported cases of SARS due to FluV were fewer in 2020 than those detected in previous years [63], the positivity of FluV and RSV generated in this cross-sectional study suggests a high possibility of under-reporting in hospitalized cases.

In an analysis stratified according to FluV subtypes, a higher prevalence of FluAV than FluBV was observed, with rates of 3% (95% CI: 2–5) and 1% (95% CI: 0–2), respectively. These data agree with the literature showing a higher incidence of FluAV than FluBV [64]. Here, the 25% FluBV infection burden was close to the 23% positivity observed by Caini et al. [65] among all FluV cases. Indeed, the highest proportion of FluAV positivity is related to its pandemic risk, as new strains, with varying degrees of virulence, can arise from a large host range with “species jumpers” to humans and spread on a global scale. This was not observed for FluBV because of its smaller host range [66].

A greater burden of RSV has been recognized in some demographic groups, particularly in children, seniors, immunocompromised and hospitalized patients, and those with comorbidities [11,12,13]. In contrast, in our cross-sectional study, the highest prevalence of RSV occurred in young adults aged 23, 32, and 40 years. However, it was not possible to determine whether these RSV cases were related to other comorbidities/immunocompromised. Notably, few studies have explored the burden of RSV in a young adult population. To reduce this gap, Htar et al. [67] conducted a meta-analysis including articles evaluating patients with community-acquired pneumonia, acute respiratory infection, and general practice with adults aged <50 years. As a result, the pooled estimate of the proportion of RSV cases ranged from 1% to 9%. Regarding RSV subtypes A and B, despite the predominance of one, both often co-circulate [68,69]. In this sense, our data showed similar prevalences of RSV-A and -B. Data related to differences in virulence have suggested a greater virulence of RSV-A than -B [70,71]. Interestingly, in preliminary observations, the new RSV-A genotype ON1 has been associated with more severe disease in young children [72,73,74]. Owing to the limited data provided by these studies, it was not possible to perform an in-depth analysis of RSV genotypes versus disease severity. Consequently, we encourage future studies to clarify this gap.

FluV/RSV and SARS-CoV-2 co-infection events have shown a very low proportion of positivity during just over a year of the COVID-19 pandemic. Indeed, our data showed that the chance of FluV/RSV positivity in a mono-infection group (i.e., FluV-/RSV-positive SCNG patients) was two to four times higher than that in co-infection cases in the SCPG. However, owing to the limitation of raw data, it was not possible to perform an in-depth analysis of whether there were differences in laboratory-clinical characteristics between the two groups. In this regard, controversial data have been obtained in other studies. For instance, a meta-analysis that focused on the clinical severity of COVID-19 patients with FluV/RSV co-infection showed a positive association between FluV and SARS-CoV-2 only in terms of a higher risk of intensive care unit admission and the need for mechanical ventilation [20]. However, Krumbein et al. [21] found an increased association between co-infected patients with dyspnea and fatality.

Curiously, the higher FluV and RSV positivity in the SCNG clinical samples when compared to the SCPG samples may be due to the short co-detectability window to adequately detect all co-infection cases via nucleic acid amplification assays [22]. This is corroborated by the evidence related to the increase in co-infection among SARS-CoV-2 patients from the second half of 2021, when there was a relaxation of non-pharmaceutical interventions (NPIs) (i.e., social distancing, respiratory etiquette, and hand hygiene) and increased circulation of non-SARS-CoV-2 respiratory viruses [75].

Another possible explanation for the low co-infection positivity among SARS-CoV-2 and other respiratory viruses (i.e., FluV/RSV) could be related to negative-type viral interactions. In this scenario, the cells infected by one virus would reduce the infection and replication of another virus. More specifically, the interferon response induced by a prior viral infection would be crucial in an antiviral state, hindering the establishment of infection and replication of a second virus. Interestingly, Fage et al. observed that during simultaneous infection, SARS-CoV-2 interfered with the replication of RSV-A2a in nasal human airway epithelium cells [76]. This was not observed for FluAV (H1N1)pdm09 virus. The authors also observed that the prior infection of this cell type by SARS-CoV-2 reduced the replication kinetics of both viruses. Pizzorno et al. explored additional key factors in virus–virus interactions and demonstrated that superinfection with H1N1 slightly reduced SARS-CoV-2 infection, and superinfection with SARS-CoV-2 had a substantial impact on primary RSV infection [77]. These findings reveal an essentially relevant point in the context of the interaction between other respiratory viruses and SARS-CoV-2.

Flu vaccine coverage and NPIs, which tended to be more stringent during the early part of the COVID-19 pandemic, considerably mitigated the incidence of FluV and influenced our pooled estimate of prevalence [78]. Therefore, it is evident that changes in NPIs, including implementation dates, duration, and the rigor of following up on such measures, considerably affected the respiratory virus positivity rate. Globally, the 2020/2021 seasonal outbreak/epidemic was drastically reduced (i.e., rhinovirus, RSV, and seasonal human CoV) [79]. However, the incidence of viral seasonal outbreaks and epidemics increased in the period mid-2021/2022 owing to a reduction in NPIs [8,80,81]. Unfortunately, owing to data limitations, it was not possible to verify whether there was still a reduced trend of co-infection with SARS-CoV-2 for the group of patients from the second half of 2021 and 2022.

Although this study is relevant, including the fact that, to the best of our knowledge, this is the first meta-analysis focusing on the burden of FluV/RSV in patients who tested negative for SARS-CoV-2, we emphasize that there are limitations. First, a high degree of heterogeneity was observed across most studies. Thus, despite well-defined criteria for selecting studies to compose the review, there were variables that could influence the analyzed population, including a wide range of ages, sex ratios, collection sites, vaccination status, and collection periods. All these variables tend to be factors that contribute to the diversity of the results. Second, for several studies, it was not possible to access the raw data, thus precluding the analysis of potential risk variables associated with viral positivity, such as FluV and RSV seasonality, the vaccination status of individuals, as well as other demographic data such as sex, age, and the presence of comorbidities in the population. Third, articles published in languages other than English, Spanish, or Portuguese were not included, with the possibility of publication bias. Fourth, the influenza vaccine coverage data were not available, which may imply a bias in the comparison between the SCNG and SCPG.

## 5. Conclusions

In summary, our study revealed a low FluV and RSV prevalence rate of 2% and 0.27%, respectively, among SCNG patients. Regarding the meta-analysis, the pooled prevalence of 4% for FluV and 2% for RSV in the SCNG was relatively low. However, the prevalence was significantly higher than that of the tested SCPG. The results of this study and continued molecular surveillance across different regions in the coming months will help to inform and deepen our understanding of the effects of viral interactions among the non-SARS-CoV-2 viral infections versus SARS-CoV-2 positivity concerning the epidemiological profile of both viruses.

## Figures and Tables

**Figure 1 viruses-15-00665-f001:**
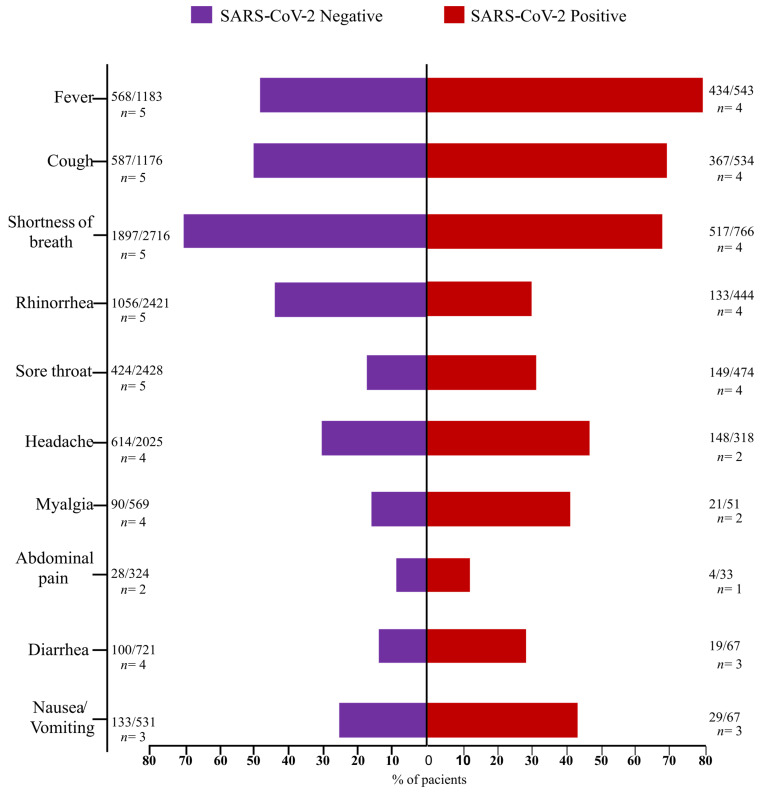
Clinical features among SCNG versus SCPG patients [46,47,51,54,59].

**Figure 2 viruses-15-00665-f002:**
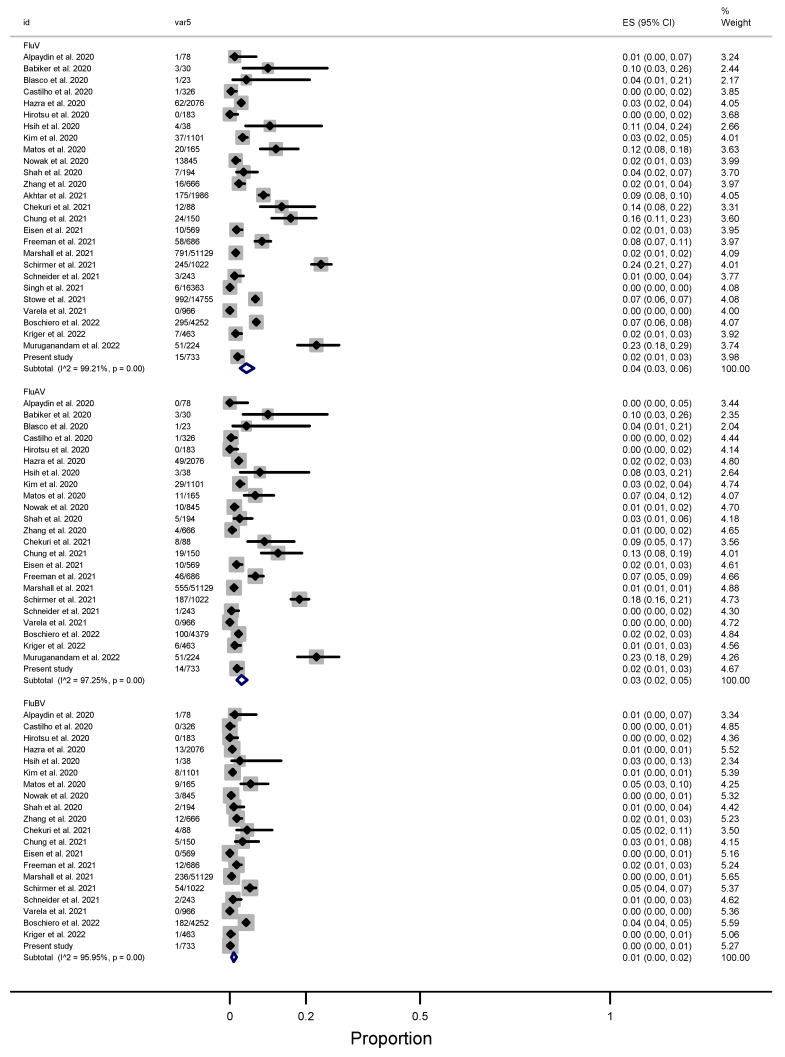
Forest plot with FluV, FluAV, and FluBV positivity among SCNG patients. The black diamonds in gray squares indicate the mean of the ratio of viral positivity, while the size of the grey square represents the weight (population size) contributed by each study in the meta-analysis. The horizontal lines in black represent their 95% CI. The blue diamond represents the pooled ratio of positives and its 95% CI. Id = identification of the study; Var5 = FluV positivity/total; ES = effect size [36,37,38,39,40,41,42,43,44,45,46,47,48,49,50,52,53,54,55,56,57,58,59,60,61,62].

**Figure 3 viruses-15-00665-f003:**
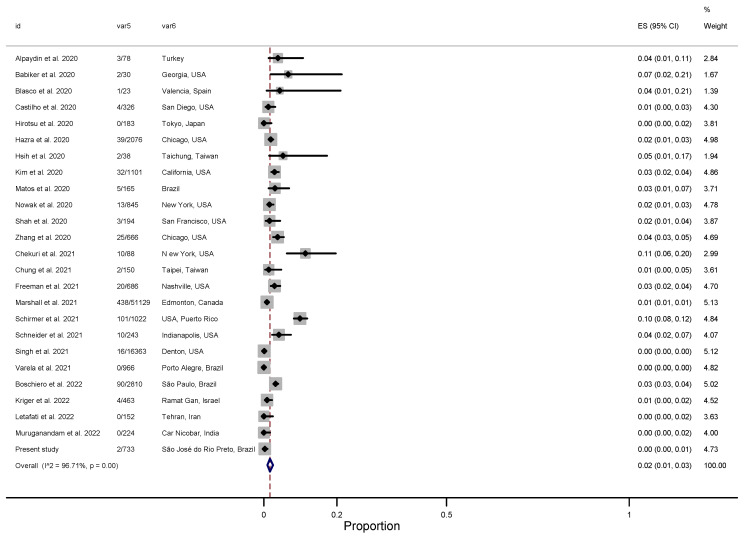
Forest plot with RSV positivity among SCNG patients. Id = identification of the study; Var5 = RSV positivity/Total; Var6 = place of study; ES = effect size [36,37,38,39,40,41,42,43,44,45,46,48,50,51,52,53,54,55,57,58,59,60,61,62].

**Figure 4 viruses-15-00665-f004:**
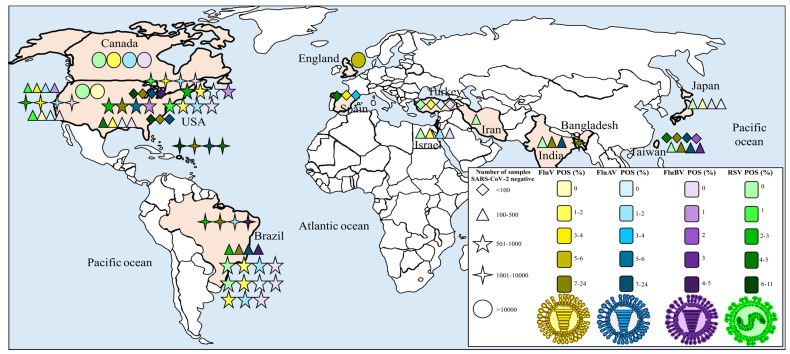
Map illustrating the geographical distribution of the included studies containing positivity values for FluV and RSV.

**Table 1 viruses-15-00665-t001:** Demographic characteristics of the study population according to the positivity of SARS-CoV-2, FluV, and RSV among suspected COVID-19 cases.

Var.	S.s	COVID+	COVID-N	*p **	FluV+/COVID+	FluV+	FluV-N	*p **	RSV+/COVID+	RSV+	RSV-N	*p **
	n (%)	n (%)	n (%)		n (%)	n (%)	n (%)		n (%)	n (%)	n (%)	
n (%)	901 (100)	168 (18.65)	733 (81.33)		3 (1.7)	18 (1.9)	883 (98)		3 (1.7)	5 (0.5)	896 (99)	
**Sex**												
Male	380 (42.3)	77 (46.67)	303 (41.33)	0.2	1 (50)	4 (24)	376 (42.7)	0.11	1 (33.3)	1 (20)	379 (42.4)	0.31
Female	518 (57.7)	88 (53.33)	430 (58.67)		1 (50)	13 (76)	505 (57.3)		2 (66.6)	4 (80)	514 (57.6)	
**Age (Y.)**												
0–4	53 (5.9)	8 (4.8)	45 (6.2)	0.5	0 (0)	1 (5.5)	52 (5.9)	0.99	0 (0)	1 (20)	52 (5.9)	0.17
5–9	6 (0.7)	1 (0.6)	5 (0.7)	0.77	0 (0)	0 (0.0)	6 (0.7)	0.73	0 (0)	0 (0)	6 (0.7)	0.85
10–19	53 (5.9)	2 (1.2)	51 (7)	0.001	0 (0)	2 (11)	51 (5.8)	0.29	0 (0)	0 (0)	53 (6)	0.57
20–29	174 (19.61)	41 (24.8)	133 (18.4)	0.4	2 (66.7)	6 (33)	168 (19.3)	0.09	1 (33.3)	1 (20)	173 (19.6)	0.96
30–39	206 (23.2)	40 (24.2)	166 (23)	0.35	0 (0)	3 (16.6)	203 (23.3)	0.62	1 (33.3)	1 (20)	205 (23.2)	0.87
40–49	178 (20)	39 (23.6)	139 (19.2)	0.93	1 (33.3)	5 (27.7)	173 (19.9)	0.23	1 (33.3)	1 (20)	177 (20)	0.98
50–59	104 (11.7)	20 (12)	84 (11.6)	0.54	0 (0)	0 (0.0)	104 (12)	0.13	0 (0)	0 (0)	104 (11.7)	0.41
60–69	26 (2.9)	9 (5.5)	17 (2.3)	0.09	0 (0)	0 (0.0)	26 (3)	0.47	0 (0)	0 (0)	26 (2.9)	0.69
70–79	19 (2.1)	0 (0.0)	19 (2.6)	0.03	0 (0)	0 (0.0)	19 (2.2)	0.54	0 (0)	0 (0)	19 (2.1)	0.74
80+	7 (0.8)	1 (0.6)	6 (0.8)	0.7	0 (0)	0 (0.0)	7 (0.8)	0.71	0 (0)	0 (0)	7 (0.8)	0.84
**C.p.**												
May/2020	697 (78.1)	134 (81.7)	563 (77.3)	0.4	3 (100)	12 (66)	685 (78.3)	0.27	3 (100)	5 (100)	692 (78)	0.22
Jun/2020	34 (3.8)	1 (0.6)	33 (4.5)	0.01	0 (0)	1 (5.5)	33 (3.7)	0.68	0 (0)	0 (0)	34 (3.8)	0.65
Jul/2020	44 (4.9)	7 (4.2)	37 (0.5)	0.6	0 (0)	4 (22)	40 (4.5)	<0.001	0 (0)	0 (0)	44 (4.9)	0.61
Aug/2020	118 (13.2)	25 (15.2)	93 (12.7)	0.45	0 (0)	1 (5.5)	117 (13.3)	0.33	0 (0)	0 (0)	118 (13.3)	0.38
Sep/2020	7 (0.8)	1 (0.6)	6 (0.8)	0.76	0 (0)	0 (0.0)	7 (0.8)	0.7	0 (0)	0 (0)	7 (0.8)	0.84

Legend: Var, variable; Y, year, C.p., collection period; S.s, sample size; COVID+, COVID-19 positive; COVID-N, COVID-19 negative; FluV+, influenza virus positive; FluV-N, influenza virus negative, RSV+, respiratory syncytial virus positive; RSV-N, RSV negative; * χ^2^ test.

## Data Availability

Not applicable.

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
