# Peer review of "Burden of Influenza and Respiratory Syncytial Viruses in Suspected COVID-19 Patients: A Cross-Sectional and Meta-Analysis Study"

_viruses, 2023, doi:10.3390/v15030665_

Round 1

Reviewer 1 Report

This is an interesting manuscript about the contribution of non-SARS-CoV-2 respiratory viral infections such as influenza virus (FluV) and human respiratory syncytial virus (RSV), in the pandemic years. The idea of this study is original and well written but the manuscript has to been improved.

The paper is a cross-sectional study in a short period fro May to September 2020, that is associated to a meta-analysis that it seems to include different paper selected by topics.

The aim of the study was to determinate the prevalence of FluV and RSV among SARS-CoV-2-negative group (SCNG) individuals.

The manuscript is well written and after some major revisions could be suitable for publication.

Although the overall conclusion reached by the authors is reasonable, as currently presented, this manuscript has a number of deficiencies that need to be addressed, outlined below: There are numerous errors of “Tense and Grammar” throughout the manuscript. Therefore, a diligent editing is in order to fix the ENGLISH language.

Major comments

1. Abstract. Please focus the abstract on your study and your results. In particular the last two sentence are vague. I will put some corrections in the minor comments.

2.More generally I suggest to focus the cross-sectional study so the specimen collection and processing and the systematic review and meta-analysis on the same pandemic period.

3. Methods. Mention the year of publications that you take in consideration for your systematic review.

4.Results. Some figures should be added to the text.

4. line are missing so doing minor comments is really difficult.

Minor comments

Abstract

Line 2: Please consider writing “ have contributed considerably” instead of the present sentence.

Line 5: Please consider writing “ remain unclear” instead of the present sentence.

Line 6: Please consider writing “and we collect the data” instead of the present sentence.

Line 6: Please consider writing “aime to evaluete” instead of the present sentence.

Line 15: Consider writing “data have shown” instead of “ the generated data”.

Line 16: Consider writing “dissemination” instead of “ circulation”.

Line 17: Please explain better this sentence.

Introduction

Line 3: Delete “one such group of pathogens” and start the sentence “Severe acute respiratory syndrome coronvirus 2 (SARS-CoV-2) etiologic agent”

Line 4: Delete “from a pneumonia outbreak” and write “was a new outbreak emerged in”

Line 14: Please consider writing “ pathogen also frequently cause of ” instead of the present sentence.

Line 15: Please consider writing “in children, senior, or immunocompromised patients” instead of the present sentence.

Line 20-21 Please consider rewriting this sentence.

Line 51: Please explain better this sentence.

Line 56-57: Please explain better this sentence.

Matherials and methods

Line 4: Delete “the”

Line 7: Please consider writing “precense of syntoms like fever or acute respiratoy discomfort” instead of the present sentence.

Line 20-21: Please consider writing “were used for the analysis between groups” instead of the present sentence.

Results

Line 1: Please consider writing “suspected COVI-19 cases” instead of the present sentence.

Line 2: Please consider writing “such as” instead of the present sentence.

Line 5-6: Please explain better this sentence.

Line 11-23 Consider explaining better all these sentences. The figure should be included in this part.

Conclusions

Not clear the conclusions.

Reviewer 2 Report

The authors carried out meta-analysis of publications and provided their own original data on the molecular epidemiology of influenza, SARS-CoV-2 and RSV infections. This manuscript is of interest to the scientific community. However, in my opinion, in the Introduction, the section on the classification of these viruses not directly related to the research topic and should be shortened or deleted. The attention should be focused on poorly studied issue, what was the basis for this work.
